# Consequences and Management of Canine Brachycephaly in Veterinary Practice: Perspectives from Australian Veterinarians and Veterinary Specialists

**DOI:** 10.3390/ani9010003

**Published:** 2018-12-21

**Authors:** Anne Fawcett, Vanessa Barrs, Magdoline Awad, Georgina Child, Laurencie Brunel, Erin Mooney, Fernando Martinez-Taboada, Beth McDonald, Paul McGreevy

**Affiliations:** 1Sydney School of Veterinary Science, Faculty of Science, University of Sydney, Camperdown, NSW 2006, Australia; vanessa.barrs@sydney.edu.au (V.B.); georginachild@gmail.com (G.C.); laurencie.brunel@sydney.edu.au (L.B.); erin.mooney@sydney.edu.au (E.M.); fernando.martinez@sydney.edu.au (F.M.-T.); beth.mcdonald@sydney.edu.au (B.M.); paul.mcgreevy@sydney.edu.au (P.M.); 2PetSure, 465 Victoria Avenue, Chatswood, NSW 2067, Australia; magdolinea@petsure.com.au

**Keywords:** brachycephalic, dogs, airways, welfare, health, veterinary ethics

## Abstract

**Simple Summary:**

Canine and human co-evolution have disclosed remarkable morphological plasticity in dogs. Brachycephalic dog breeds are increasing in popularity, despite them suffering from well-documented conformation-related health problems. This has implications for the veterinary caseloads of the future. Whether the recent selection of dogs with progressively shorter and wider skulls has reached physiological limits is controversial. The health problems and short life expectancies of dogs with extremely short skulls suggests that we may have even exceeded these limits. Veterinarians have a professional and moral obligation to prevent and minimise the negative health and welfare impacts of extreme morphology and inherited disorders, and they must address brachycephalic obstructive airway syndrome (BOAS) not only at the level of the patient, but also as a systemic welfare problem.

**Abstract:**

This article, written by veterinarians whose caseloads include brachycephalic dogs, argues that there is now widespread evidence documenting a link between extreme brachycephalic phenotypes and chronic disease, which compromises canine welfare. This paper is divided into nine sections exploring the breadth of the impact of brachycephaly on the incidence of disease, as indicated by pet insurance claims data from an Australian pet insurance provider, the stabilization of respiratory distress associated with brachycephalic obstructive airway syndrome (BOAS), challenges associated with sedation and the anaesthesia of patients with BOAS; effects of brachycephaly on the brain and associated neurological conditions, dermatological conditions associated with brachycephalic breeds, and other conditions, including ophthalmic and orthopedic conditions, and behavioural consequences of brachycephaly. In the light of this information, we discuss the ethical challenges that are associated with brachycephalic breeds, and the role of the veterinarian. In summary, dogs with BOAS do not enjoy freedom from discomfort, nor freedom from pain, injury, and disease, and they do not enjoy the freedom to express normal behaviour. According to both deontological and utilitarian ethical frameworks, the breeding of dogs with BOAS cannot be justified, and further, cannot be recommended, and indeed, should be discouraged by veterinarians.

## 1. Introduction

Brachycephaly—the foreshortening or flattening of the facial skeleton—is a mutation that is selected for in some dog breeds, such as the French Bulldog, British Bulldog and Pug [1]. Since the BBC television program *Pedigree Dogs Exposed* aired in the UK in 2008, highlighting breed-related welfare issues, there has been media and scholarly attention on the welfare problems associated with brachycephalic breeds. Despite well-documented conformation-related health problems, brachycephalic dog breeds are increasing in popularity.

Since 2007, the UK Kennel Club has reported a 3104% increase in French Bulldog registrations, a 193% increase in Pug registrations and a 96% increase in Bulldog registrations [2]. These trends are echoed in Australia where, since the mid-1980s, consumers have gradually favoured shorter, smaller, brachycephalic breeds, with the weighted mean cephalic index increasing from 57.7 in 1986 to 62.9 in 2013 [3]. According to figures published by the Australian National Kennel Club (ANKC), there was an 11.3% increase in French Bulldog registrations (as a proportion of total purebred registrations), a 320% increase in Pug registrations, and a 324% increase in British Bulldog registrations between 1987 and 2017 [4].

There is now substantial evidence that brachycephaly compromises the welfare of affected dogs by leading to brachycephalic obstructive airway syndrome (BOAS) [1]. BOAS is a progressive disorder of the upper airway that is characterized by primary anatomical abnormalities (stenotic nares, an elongated soft palate, distortion of pharyngeal soft tissues, and a hypoplastic trachea) causing resistance to airflow and restricted breathing. This leads to increased negative intraluminal pressure during inspiration, exacerbating soft tissue abnormalities, leading to everted laryngeal saccules, pharyngeal hyperplasia, tonsillar hyperplasia, and eventual laryngeal collapse [5]. Laryngeal collapse has been reported in dogs as young as 4.5 months [6].

Clinical signs of BOAS include stertor, stridor, dyspnoea, exercise intolerance, gagging, regurgitation, cyanosis, hyperthermia and syncope [1]. Extreme brachycephalic breeds (Bulldog, French Bulldog, and Pug) were 3.5 times more likely to suffer from at least one upper respiratory tract (URT) disorder than moderate or non-brachycephalic breeds, and were more likely to die due to a URT disorder [7].

Dogs with BOAS may suffer from concurrent disorders that are unrelated to their upper airway abnormalities, including ophthalmic conditions, spinal malformation and dystocia [8]. Extreme brachycephalic dogs had shorter life spans (median longevity 8.6 years) when compared with moderate and non-brachycephalic dogs (median longevity 12.7 years) [7]. A fundamental consequence of airway turbulence is the reduced clearance of carbon dioxide, which can lead to acidosis [9]. How this, along with hypoxia and hypertension, affects enzyme function and oxidative stress is unclear. It is possible that shifts in homeostasis may have effects on longevity.

In a survey of 15 national kennel clubs, exaggerated morphological features and inherited disorders and inherited disorders were identified as the most pressing concerns, with kennel clubs expressing a strong desire to improve the breeding and health of pedigree dogs, including brachycephalic breeds [10]. In addition to veterinary professional organisations and animal welfare charities, other organisations are working to address these concerns. For example, in the UK, the Advisory Council on the Welfare Issues of Dog Breeding was formed to address inherited disorders that impact canine quality of life, including extreme brachycephalism and disorders like syringomyelia [11]. The International Partnership for Dogs, a not-for-profit organization registered in Sweden, is working with kennel clubs, professional groups and individuals to improve the health, well-being and welfare of brachycephalic breeds [12]. In addition, kennel clubs are increasingly investigating fraud in the breeding and registration of brachycephalic breeds. For example, in 2017, the ANKC established a French Bulldogs taskforce to investigate fraudulent breeding and registration practices in French Bulldogs, British Bulldogs, Pugs and Staffordshire Bull Terriers [13].

The current article explores the implications of extreme brachycephaly for affected animals, from the perspectives of veterinarians and veterinary specialists treating these animals. It provides a review of the known health conditions, welfare impacts and challenges for patient management, to underscore the enormity of the welfare problems posed by extreme brachycephaly (Section 2, Section 3, Section 4, Section 5, Section 6, Section 7 and Section 8). Ultimately, we argue that veterinarians are ethically obliged to address BOAS, not only at the level of the patient, but also as a systemic welfare problem.

## 2. Australian Insurance Data

A recent survey estimated that approximately 609,000 Australian companion animal owners had a pet insurance policy, representing a 6% overall insurance rate for cats and dogs [14]. Insurance rates for dogs are higher than for cats. Of the estimated 5.4 million Australians who have a dog, 7.1% (or approximately 383,400) have pet insurance. This is likely an underestimate. PetSure Australia (Sydney, Australia) is the largest underwriter of pet insurance in Australia, and at the time of publication, it underwrites 391,791 canine insurance policies across 28 brands (Petsure Australia 2018, unpublished data).

Petsure Australia has collected relevant prevalent and actuarial data over the years, in the process of administering its core business of providing pet insurance policies. This includes the following brachycephalic breeds from their database in its ongoing in-house study: Affenpinscher, American Bulldog, Australian Bulldog, Australian Bulldog Miniature, Boston Terrier, Boxer, British Bulldog, Cavalier King Charles Spaniel (CKCS), Dogue De Bordeaux, French Bulldog, Griffon, Griffon Brabancon, Griffon Bruxellois, Lhasa Apso, Mastiff, Neopolitan Mastiff, Pekingese, Pug and Shih Tzu. PetSure’s data confirm that there has been a steady rise in the popularity of French Bulldogs over the past five years (Figure 1).

Pugs have also increased in popularity over the same period, and so the rise in the popularity of both breeds has contributed significantly to the proportion of brachycephalic breeds insured Table 1. The ownership rate of the British Bulldog has almost tripled in Australia since 2013. However, the percentage of insured boxers, a traditionally popular brachycephalic breed, has remained steady over the same period.

Importantly, data from PetSure demonstrate the health complications that are associated with severe canine brachycephalism. There was a 69% rise in BOAS as a proportion of all brachycephalic dog surgery claims between 2013 and 2017. A possible explanation is the increase in the proportion of French Bulldogs, Pugs and British Bulldogs, breeds that have more exaggerated facial features, requiring surgical intervention. However, this may also be due to other factors, including an increased awareness of welfare concerns associated with BOAS among veterinarians, training of general practitioners in surgical treatment of BOAS, increased capacity to refer to surgical specialists and an increase of dogs that are covered by insurance.

Many veterinarians combine de-sexing with “other surgeries/procedures”. While it is not unusual in practice to repair umbilical hernias or to remove deciduous canine teeth at the time of desexing (traditionally performed at about six months of age), PetSure data from 2013–2016 reveal a difference in the rate of “other surgery” performed at the time of de-sexing in 11% of brachycephalic dogs, compared to 7% of non-brachycephalic dogs. The most common procedures performed were hernia repairs, BOAS surgery and cherry eye repair, in that order.

While much is known about the incidence of airway disease in brachycephalic dogs, many other conditions are also more likely to occur in these breeds at incident rates that are greater than in other breeds. Figure 2 shows that prevalence of insurance claims relating to common diseases in brachycephalic and non-brachycephalic breeds. It is worth noting that not all PetSure policies cover dental disease, and no polices include coverage for breeding or obstetric procedures, therefore conditions such as caesarean sections (which are commonly performed in brachycephalic breeds), do not appear in the Figure 2 below.

Data collected from PetSure’s database show that whilst brachycephalic breeds have an increased incidence of diseases related to their facial conformation, they are also more likely than other breeds to be affected by a range of other common diseases. The prevalence of BOAS and the incidence of other common diseases is also much higher in pure-bred brachycephalic dogs when compared to their crosses. These findings are in line with a previous analysis of analysis of over 1.27 million pet insurance claims over a nine-year period (2007–2015), excluding conditions specifically linked to brachycephaly (stenotic nares, elongated soft palate, tracheal hypoplasia and everted laryngeal saccules), found that brachycephalic dogs were more likely to suffer from a range of conditions when compared with non-brachycephalic dogs [15]. These included, but were not limited to, corneal ulceration, ocular trauma, heat stroke, pneumonia, fungal skin disease and skin cancer.

Such information is not only valuable for pet insurance companies, but also for the veterinarians who treat these conditions, as well as owners of at-risk breeds, who are likely to incur significant financial costs over the lifespan of these dogs.

## 3. Stabilization of Respiratory Distress

For dogs, panting is an important component of heat dissipation. Patients with brachycephaly have greater respiratory resistance to airflow and a reduced ability to thermoregulate [5]. As such, heat stress and upper airway obstruction are intrinsically linked. 

Because of upper airway obstruction, brachycephalic patients must work harder to breathe. Upper airway obstruction is common in patients with brachycephaly. The precipitating causes include intrinsic end-stage airway collapse and swelling of the airways caused by panting or tachypnoea, or following sedation or anaesthetic, when the airway musculature is relaxed. When their respiratory drive increases due to hyperthermia, anxiety, pulmonary disease, or other causes, excessive negative pressures need to be generated to move larger volumes of air. As a consequence, the soft tissues of the upper airway swell, further compromising airflow and heat dissipation. A vicious cycle ensues; heat stress can lead to upper airway obstruction, and vice versa.

### 3.1. Intubation

Brachycephalic patients can present with either partial or complete airway obstruction. Patients undergoing a partial obstruction will be dyspnoeic, with an audible stertor or stridor, typically with paradoxical motion of the abdomen and orthopnoea. Those with complete obstruction may not generate any sounds at all as they are unable to move air.

Upon presentation, the clinician must decide on whether to intubate the patient, depending on the severity of the obstruction. Patients with complete airway obstruction and in respiratory arrest, or for whom the effort of breathing appears unsustainable should be intubated immediately. If there is time, a quick airway examination should be performed so that the primary cause of obstruction can be identified before intubation. For all but arrested patients, intubation will require anaesthetic induction; an intravenous (IV) catheter should be placed as quickly as possible, and a short-acting induction agent, such as propofol or alfaxan, administered (typically without pre-medication) to effect. If the patient cannot be intubated, a temporary tracheostomy should be placed. 

### 3.2. Thermoregulation

For patients without complete airway obstruction, conservative management can be first attempted. However, patients should not be left unsupervised until they are stable, as intubation can become necessary at any time. Conservative management revolves around three principles: oxygen supplementation, sedation, and if the patient is hyperthermic, active cooling. It is imperative that these patients are managed with as little stress as possible, because struggling increases oxygen demand, and this can precipitate respiratory arrest. Oxygen can be provided via flow-by, mask (although this can be stressful), oxygen hood, or oxygen cage. Sedation can be administered intramuscularly, and an IV catheter placed when the patient is more stable. Some patients will tolerate a IV catheter placement immediately. Butorphanol (0.2–0.3 mg/kg IV ± medetomidine 5 mcg/kg IV) is a reliable protocol. The goal of sedation is to reduce anxiety and slow the respiratory rate, which will reduce turbulence and improve airflow. The patient should remain conscious and retain a gag reflex. If the gag reflex is lost, the patient must be intubated to protect the airways from aspiration, which can lead to pneumonia. A dose of corticosteroids (dexamethasone 0.15 mg/kg IV or IM) can be administered to help reduce airway swelling. Corticosteroids should not be given to patients during heat stress, due to potential gastrointestinal tract (GIT) complications such as GIT ulceration. 

All patients should have their rectal temperature taken as soon as is reasonable. Those with temperature greater than 39.5 °C should receive active cooling as follows: administer room temperature IV fluids, wet the hair-coat with room temperature tap water or wet towels, and place a fan in front of the patient. The rectal temperature should be checked every 10 mins. Active cooling should be stopped once the temperature reaches 39.4 °C to prevent rebound hypothermia [16].

Patients with heat stress should also be screened for organ injury. The GIT is the most sensitive to damage, manifesting as vomiting, diarrhoea (often haemorrhagic) and ileus. This can be a major source of fluid loss. Other potential complications include acute kidney injury, coagulopathy and hypoglycaemia [17]. Altered mentation is also common, but it is typically quick to resolve.

### 3.3. Complications

Patients with upper airway obstruction commonly succumb to pulmonary complications such as aspiration pneumonia/pneumonitis and non-cardiogenic pulmonary oedema. Once the patient is stable, it is important to assess pulmonary function. Parameters such as respiratory rate and effort, thoracic auscultation, pulse oximetry and arterial blood gas analysis can all be used to assess pulmonary function. If there are any concerns, thoracic radiographs should be performed. The appearance of aspiration pneumonia on radiographs is typified by an alveolar pattern in the cranioventral lung fields, whereas non-cardiogenic pulmonary oedema tends to organize in the caudodorsal lung fields. However, some radiographic signs overlap. Patients who are suspected of having pulmonary injury should be treated with supplementary oxygen. If aspiration pneumonia is suspected, empirical antibiotic therapy should be provided.

### 3.4. Extubation

Successful extubation of patients with upper airway obstruction is difficult, as swelling typically persists for a few days, and anaesthesia reduces tone in the airway muscles. It should only be attempted once the patient is normothermic, and after diagnostics have been performed, and most patients will require one or more corrective airway surgeries before they can be extubated successfully. For hypoxaemic patients, the clinician should have a plan in place for providing supplemental oxygen prior to extubation. This can pose a challenge in brachycephalic patients due to their facial conformation, and naso-tracheal catheters should be considered [18].

Extubated patients must not be left unattended for the first 12 h, as many will require emergency re-intubation. If transfer to a 24-h facility is required, it is safest to transport the patient while they are intubated. Alternatively, a temporary tracheostomy can be placed and the patient recovered from anaesthetic. In a review of 42 cases where dogs underwent temporary tube tracheostomy, the main indication was BOAS. Although 81% of dogs survived until tube removal, 86% had complications [19]. The most frequent complications included tube obstruction, dislodgement, aspiration pneumonia, stoma swelling and bradycardia association with tube manipulation and cleaning.

### 3.5. Monitoring

Most patients with BOAS presenting with respiratory distress should ultimately be referred for an airway examination and possible corrective surgery. The exception are those patients who have a clear initiating cause for their airway obstruction such as severe environmental heat stress, which can be prevented from occurring again. With or without surgery, owners should be educated on how to limit the chances of another obstructive episode. Patients should not be exercised on warm or humid days, should be kept in a cool area of the house in summer months, and be subject to veterinary supervision immediately if their breathing seems to be more laboured than usual. 

## 4. Sedation and Anaesthesia 

The booming popularity of brachycephalic breeds is making the challenge of sedating and anaesthetizing them reasonably common in daily veterinary practice. It is important for both veterinarians and owners to appreciate the risks that are associated with the sedation and anaesthetization of brachycephalic patients. Regardless of whether they present with airway issues (e.g., laryngeal examination and tachypnoea) or an unrelated presentation (e.g., intradermal testing, caesarean sections or stifle surgery), the general approach to sedation and anaesthesia in these breeds is similar.

### 4.1. Peri-Anaesthetic Morbidity and Mortality

Although there is an overall impression among veterinary anaesthetists that brachycephalic breeds are predisposed to a greater mortality risk from anaesthesia, no published studies support this. This might reflect differences in study populations among the different countries where data was collected, and the periods during data was collected. However, the data does show that for brachycephalic dogs undergoing surgery for BOAS, 87% have stertor or stridor, 22% have GIT signs at presentation (vomiting or regurgitation), and 15% have had previous BOAS surgery performed [20]. These data also indicate that 7% of BOAS patients encounter major post-operative complications (requiring a tracheostomy tube and/or resulting in death or euthanasia) characterised by postoperative radiographic evidence of pneumonia, and associated with the perioperative use of metoclopramide in infusion.

### 4.2. Anaesthetic Considerations

Taking into consideration the anatomical attributes of the BOAS and the common comorbidities, the most important considerations at the time of anaesthesia in brachycephalic dogs are upper respiratory obstruction and poor oxygenation, poor ventilation and anaesthetic uptake, regurgitation, agitated recovery and postoperative inflammation.

#### 4.2.1. Upper Respiratory Obstruction and Poor Oxygenation 

Most of the anatomical features of BOAS relate to upper respiratory obstruction, and tracheal intubation can relieve this. Intramuscular sedation is recommended to calm the dog before attempting to place an IV catheter. A combination of acepromazine (0.01–0.02 mg/kg), medetomidine (5–10 µg/kg), and an opioid produces moderate sedation, with the advantage of facilitating lower doses of sedative drugs, having a long-lasting effect from the acepromazine, and the possibility of reversing the effect from the medetomidine at the end of the procedure (to minimise the time to recovery). Pre-oxygenation is essential in these animals, as tracheal intubation can be difficult, especially in cases of everted laryngeal saccules or secondary laryngeal paralysis. The induction of anaesthesia by using fast-acting drugs (e.g. propofol and alfaxalone) is recommended.

#### 4.2.2. Poor Ventilation and Anaesthetic Uptake

Brachycephalic dogs often have some degree of respiratory depression, characterized by a lower tidal volume and a fast respiratory rate. The decreased tidal volume directly limits alveolar gas, so that respiratory acidosis is common. Under anaesthesia, the pattern of ventilation remains very similar, with the addition of further respiratory depression and muscle relaxation from the anaesthetic drugs. As a result, the uptake of volatile anaesthetics is decreased, and the use of intermittent positive pressure ventilation (IPPV) is recommended during anaesthesia.

#### 4.2.3. Regurgitation

GIT signs are extremely common in brachycephalic dogs [21]. Short starvation periods are recommended, because they are associated with lower regurgitation rates (lower cardiac sphincter tone relates to stomach acidity) [22]. The administration of omeprazole [23] and metoclopramide [24] is also associated with reduced regurgitation. Maropitant reduces vomiting in dogs, but has no effect on regurgitation [25]. For elective surgery, it is advisable to commence administration of omeprazole orally the day before the procedure, to prevent regurgitation, and to administer a metoclopramide infusion as soon as clinical signs appear. 

#### 4.2.4. Agitated Recovery and Postoperative Inflammation

The recovery period is as critical as the period between premedication and tracheal intubation. The dog must be allowed to recover in a calm environment. Extubation is normally delayed until the dog is fully awake and cannot tolerate the tube any longer, to maximize airway patency and oxygenation. Unless contraindicated, steroids are often administered during the procedure, or immediately afterwards, to minimize inflammation related to tracheal intubation and any surgical correction performed in the larynx. Supplemental sedation or oxygenation might be required to support peripheral perfusion and oxygenation. Monitoring of the haemoglobin oxygen saturation via a pulse oximeter, and the assessment of the respiratory rate and effort are essential, because approximately 50% of cases of peri-anaesthetic mortality in dogs occur during recovery [26].

## 5. Surgical Treatment of Airway Abnormalities 

As brachycephalic breeds have become very popular, respiratory distress associated with BOAS is commonly seen in veterinary practice. Moreover, selection for brachycephalic phenotypes has worsened the upper airway stenosis, and dogs are consequently affected at a younger age [27].

Early intervention and correction of BOAS abnormalities are recommended for the decrease or prevention of the progression of airway pathology [28]. The typical short, wide skull leads to compression of the nasal passages, as well as the obstruction of the nasopharyngeal area by aberrant turbinates [29,30]. Recent studies using computed tomographic (CT) evaluation of airway dimensions have shown that Pugs are particularly affected by these skull changes, with the dorsal rotation of the maxillary bone, whereas French Bulldogs have more severe soft tissues changes, especially the thickening of the soft palate [31,32]. Primary soft tissue changes include stenotic nares, an elongated and thickened soft palate, nasopharyngeal mucosal hyperplasia, everted tonsils and macroglossia. Laryngeal collapse occurs secondary to turbulent airflow and chronic high negative pressures in the pharynx. The severity of laryngeal collapse is graded from Stage 1 to Stage 3. Everted laryngeal saccules are considered typical of Stage 1. Stage 2 is characterized by a medial displacement of the cuneiform processes and Stage 3 by a collapse of the corniculate processes of the arytenoid cartilages [30,33].

Dysphagia, vomiting and regurgitation are common clinical signs in brachycephalic breeds. The prevalence of hiatal hernia and oesophageal disease is high in French Bulldogs, and investigation for the evidence of gastroesophageal disease should be part of the BOAS diagnostic investigation prior to treatment [34,35].

### 5.1. Surgical Therapy

Surgical correction of BOAS requires the patient to be positioned in sternal recumbency, with the chin resting on a well-padded surface. The maxilla should be suspended by tape hung between two infusion poles or similar structures. The cuff of the endotracheal tube should be adequately inflated.

#### 5.1.1. Stenotic Nares

Several surgical options have been described for the correction of the stenotic nares. Alaplasty is the most common procedure, and it involves the excision of a wedge of the ala nasi, with a primary closure of the defect. The wedge excision can be made vertically, horizontally or laterally. Based on preference, a scalpel blade, laser or electrocautery can be used. Closure can be performed with a 4-0 or 5-0 monofilament suture, using two or three simple interrupted stitches. The use of a rapidly absorbable suture abrogates the need for suture removal, which can otherwise necessitate sedation. 

#### 5.1.2. Turbinectomy

Laser-assisted turbinectomy (LATE) is aimed at the removal of the aberrant and obstructive parts of the ventral and medial nasal turbinates. In preliminary reports, LATE, combined with vestibuloplasty and staphylectomy, demonstrated a 50% reduction of intranasal resistance in brachycephalic breeds [36,37,38]. 

#### 5.1.3. Hyperplastic Soft-Palate

The soft palate should be evaluated prior to intubation, with the head and tongue in a neutral position. In this position, a normal soft palate should not extend past the tip of the epiglottis, or the middle to caudal aspect of the tonsillar crypt [28,33].

The most common surgical technique is intended to shorten the soft palate by excision of its caudal portion (staphylectomy), to prevent it from obstructing the rima glottidis during inspiration. Different landmarks for incisions have been recommended, including the tip of the epiglottis, the middle to caudal aspect of the tonsils, or more recently, a CT-enabled pre-calculated distance from the pterygoid processes. During staphylectomy, the caudal border of the soft palate is prehended with an Allis forceps before resection of excessive length is performed with a scalpel blade, metzenbaum scissors, monopolar electrocoagulation, CO2 laser, diode laser, or bipolar sealing device [39].

As the staphylectomy does not address the soft palate thickening and consequent nasopharyngeal obstruction, a folded palatoplasty technique has been described to correct both the excessive length and the thickness of the soft palate. In this technique, the soft palate is made thinner by an excision of a portion of its oropharyngeal mucosa and the underlying soft tissues. The palate is then shortened by being folded onto itself until the caudal nasopharyngeal opening is visible transorally [30,40].

#### 5.1.4. Everted Laryngeal Saccules

Excision of everted laryngeal saccules remains controversial [41,42,43]. However, they should be removed when the eversion contributes significantly to the obstruction of the rima glottidis. Complications, such as laryngeal webbing and regrowth, can occur [44].

#### 5.1.5. Laryngeal Collapse

The treatment of laryngeal collapse is considered only when clinical signs do not improve with appropriate treatment of the nares and soft palate. Laser-assisted partial arytenoidectomy and arytenoid lateralization may also merit consideration, while a permanent tracheostomy may be considered palliative [41].

### 5.2. Prognosis After Surgical Therapy

Around 90 per cent of BAS dogs experience significant improvement of respiratory function with surgery [30]. However, despite appropriate surgical treatment, respiratory function remains compromised in 60% of dogs [38]. Younger age, normal body condition score, presence of laryngeal collapse and traditional multilevel surgery versus modified multilevel surgery are reported as negative prognostic factors after surgical treatment [45]. Age was positively associated with the odds of temporary tracheostomy tube placement in dogs after BOAS surgery, indicating that early identification and surgery may be beneficial [46]. Perioperative mortality rates have improved to less than 4% in recent studies [47].

## 6. Effects of Brachycephaly on the Brain and Associated Neurologic Abnormalities 

The main neurologic disorder in dogs associated with brachycephaly is syringomyelia (SM), which is most commonly accompanied by a Chiari-like malformation of the skull (CM). SM is a cavitatory accumulation of fluid within the spinal cord [48]. Syrinx formation may occur over several segments of the spinal cord, or extend to its entire length, and is prevalent in the cervical spinal cord of brachycephalic breeds. While SM can occur as a sequel to other spinal cord pathologies, it is recognised most commonly as a developmental abnormality in breeds with a shorter than normal basi-cranium (skull base), which is typical of brachycephalic and many miniature breeds.

Shortening of the basicranium is thought to reflect premature closure (synchondrosis) of the centres of endochondral ossification within the basisphenoid and basioccipital bones [49]. Shortening of the basicranial axis may lead to compensatory changes in other cranial bones, including widening and doming of the skull, and may contribute to neural overcrowding. CM (Chiari-like malformation) with abnormality of the occipital bone, resulting in compression of the contents of the caudal fossa (cerebellum and brain stem) and herniation of the cerebellum through the foramen magnum, is probably multifactorial. The contributing factors proposed include increased cerebellar volume, with failure of the caudal cranial fossa to reach a commensurate size [50], and kinking of the medulla at the cranio-cervical junction [51]. CM has been described most commonly in the CKCS, and is reported to occur in 100%of dogs of this breed [52]. SM may also be seen in dogs with a shortened basi-cranium without CM, including overtly brachycephalic breeds such as the Griffon Bruxellois [53,54] and miniature breeds, such as Pomeranians and Chihuahuas [55]. Increasingly, it is being reported in other toy breeds including Boston Terriers, Affenpinschers, Yorkshire Terriers, Pomeranians and Chihuahuas [55,56,57]. SM in all of these breeds is believed to develop, due to altered cerebrospinal fluid (CSF) flow at the craniocervical junction [48]. In miniature breeds, other cranio-cervical junction anomalies, including abnormalities of C1 and C2, may contribute to the clinical signs [55]. Affected dogs may have concurrent ventriculomegaly.

CKCSs and Griffons Bruxellois are highly predisposed to SM, with a prevalence in dogs without recognised clinical signs being estimated at 25% at 12 months of age, 70% at six years or older in CKCSs [58] and 52% in Griffons (of varying ages) [54]. The incidence of dogs showing clinical signs associated with CM and/or SM is much lower. CM and SM have a moderately high heritability in CKCSs [59], and genetic factors are likely in other affected breeds [60] and many of their crosses [61]. SM is almost certainly associated directly with skull conformation. The mode of inheritance is unclear, but it does not appear to be simple. Phenotypic screening of CKCSs and Griffons Bruxellois in the last 10 years seems to have reduced the incidence of severely affected dogs in breeding lines, where screening with magnetic resonance imaging (MRI) has been undertaken, but has not eliminated the condition [51,62].

### 6.1. Clinical Signs and Diagnosis

Some features of CM may be slowly progressive with time, and include changes in foramen magnum height, and herniation of the cerebellum [63]. Clinical signs, manifesting as episodic apparent neck pain, may be associated with CM without SM [48]. SM is progressive, and clinical signs are generally more severe, but they may not emerge until the affected animals are over four years of age. More severely affected dogs may present at a younger age and, on occasions, severe clinical signs may be seen in dogs under one year of age [48].

The most common presenting clinical signs of SM are cervical pain, which may be episodic, cervical hyperaesthesia, characterised by persistent scratching of one area of the shoulder, neck, sternum, ear or flank [48]. “Phantom” scratching often occurs when dog is moving or physically stimulated (e.g., by touching the neck), and is seen most commonly on one side of the body. This scratching is thought to be the result of abnormal sensation (allodynia), but may represent an abnormal scratch reflex [48]. Other neurological deficits include thoracic limb weakness (such that forelimbs splay laterally) and mild hindlimb ataxia. Other demonstrable neurologic deficits are uncommon, except in severe disease or in dogs with concurrent cranio-cervical junction anomalies. Regardless of the cause of SM, scoliosis (lateral deformity of the spinal column) may be evident on radiographs. A diagnosis of CM and/or SM is made after MRI of the brain and spinal cord.

### 6.2. Treatment 

In dogs, SM progression may occur over years, and sometimes it may never be associated with clinical signs. CM and/or syringomyelia is commonly found as an incidental finding in affected breeds without clinical signs during the investigation of other neurological diseases. Clinical signs are more likely to be seen in dogs with larger and asymmetric syrinxes [64]. It is not predictable which dogs will ever exhibit clinical signs associated with CM or SM.

For dogs showing clinical signs, such as spinal hyperaesthesia, persistent scratching and/or neurological deficits, low-dose prednisolone and/or treatment with proton pump inhibitors (such as omeprazole) can be used to reduce CSF production [48]. Gabapentin (or pregabalin) is often used in the medical treatment of dogs with spinal pain or persistent scratching [48]. The use of prednisolone in the longer term is reserved for dogs with signs that are unresponsive to these drugs, due to potential adverse effects, including iatrogenic hyperadrenocorticism and gastric ulceration. Non-steroidal anti-inflammatory drugs (NSAIDs) or amitriptyline can also be used treat those dogs with spinal pain as the predominant clinical sign. Surgical decompression of the foramen magnum has also been proposed to treat dogs that are unresponsive to medical management [65], although no technique has been shown to demonstrate syrinx resolution [66]. The prognosis for affected dogs is variable. However, for those with marked scoliosis, intractable spinal pain and/or neurologic deficits that are unresponsive to medical management, the prognosis is poor. 

In affected breeds, MRI screening of potential breeding animals is recommended [48]. Selecting for a non-exaggerated skull conformation (notably for a less broad, less domed, longer skull) is also recommended. Two significant risk factors that are associated with the development of CM/SM in the conformation of the CKCS are the extent of the brachycephaly, and the distribution of the cranium (higher cephalic index with a broader, shorter head and rostrocaudal doming [67]. The more the cranium (top of the skull) is distributed caudally rather than rostrally, the more dogs seem to be spared syrinx development [67]. Any change in preferred skull conformation will depend, in part, on the pet-buying public and possibly dog-show judges being educated about the health risks that are associated with the wide, flat face that has been regarded to be more appealing.

### 6.3. Other Neurological Conditions

Brachycephalic screw-tailed dogs, including the French Bulldog, Pug and British Bulldog, are commonly affected by vertebral malformations, including hemivertebra, and spinal curvature abnormalities, including kyphosis and scoliosis [68,69,70]. Multiple malformations are common. These may lead to vertebral misalignment, stenosis and instability, and potentially intervertebral disc extrusion. For example, French Bulldogs with kyphosis were almost twice as likely to suffer from intervertebral disc extrusion, particularly in the thoracolumbar region [69]. In the same study, spinal scoliosis was associated with a higher risk of caudal lumbar intervertebral disc extrusion. Clinical signs of neurologic dysfunction vary with the location of vertebral malformation. Clinical signs that are associated with thoracic vertebral malformation include paraparesis and pelvic limb ataxia, which may be acute or chronic [70]. Some dogs may be non-ambulatory. Screw-tailed breeds have an increased incidence of sacrocaudal malformations, which may be associated with urinary and/or faecal incontinence. Some dogs may require surgery to relieve spinal cord compression. Vertebral malformations including spina bifida may be associated with other neural abnormalities, including meningocoeles and dermoid sinuses, with varying clinical signs [71].

## 7. Dermatological Conditions

Selection for brachycephalic appearance and screw tail has resulted in deep skin folds and subsequent intertrigo. Consequently, brachycephalic dog breeds share a well-documented list of dermatological disorders, including facial and tail fold intertrigo, pattern baldness, atopic dermatitis, demodicosis, *Malassezia* dermatitis, mast cell tumours, muzzle and pedal folliculitis and furunculosis [72]. In addition, individual breeds have specific dermatological disorders, including primary secretory otitis media (POSM) in the Cavalier King Charles Spaniel, and flank alopecia in the English Bulldog and Boxer.

In the study by Webb Milum et al, their original aim was to identify a normal, healthy population of English Bulldogs without signs of dermatitis (pododermatitis, pruritus) or gastrointestinal disease, and to compare them with a population of English Bulldogs with dermatological disease [73]. They evaluated 34 English Bulldogs at two dog shows, and all dogs had evidence of some form of dermatitis, many had a history of facial pruritus and over 50% had been recently been medicated for dermatology conditions. The study could not be completed, as they were unable to find an English Bulldog that was free of dermatological or GIT disease.

The low genetic diversity in this breed not only perpetuates dogs that have been bred with physical abnormalities that are associated with brachycephalic conformation, but also impacts on the genes that regulate their normal immune responses [74].

Many of the dermatological diseases that commonly occur in the brachycephalic breeds require long-term management, such as allergen-specific immunotherapy for atopy, control of staphylococcal and *Malassezia* infections, both systemically and topically, surgical resection of pedal furuncles of the paws (interdigital cysts), parasiticides for demodicosis, and/or topical therapy or surgical resection of folds for facial and tail fold intertrigo.

While demodex mites are considered part of the normal skin flora in dogs, they become pathogenic when the immune system is compromised and mite numbers increase. This results in inflammation and infection. Isoxazoline parasiticides were initially designed to offer flea and tick protection, but they are now recognized (and registered in some countries) for treatment against Demodicosis [75].

In canine atopic dermatitis, clinical lesions are found on the pinnae, flexor surface of the elbows, muzzle and glabrous areas. It is often complicated by the presence of secondary *Malassezia* dermatitis and otitis, staphylococcal infections, and defects in the epidermal barrier function [72]. Treatment guidelines have been updated in 2015 by the International Committee on Allergic Diseases of Animals, ICADA [76].

*Malassezia* dermatitis is most commonly found on the paws, lip folds, ear canals, perineal area, folds and ventral neck. Lesions are characterised by pruritus, inflammation, hyperpigmentation and lichenification and a greasy exudate. Successful management requires the control of the primary disease, and a combination of systemic and topical therapy [77].

## 8. Other Conditions

Brachycephalic facial conformation can cause lagophthalmos (the inability to close the eyelids completely), and predisposes affected dogs to corneal ulceration, a painful eye condition that can result in scarring, corneal perforation or permanent blindness. In a study of 700 dogs, 31 were affected by corneal ulcers, with Pugs being the most commonly affected [78]. Brachycephalic dogs (those with a craniofacial ration of <0.5) were 20 times more likely to suffer corneal ulceration than non-brachycephalic breeds, while those with a 10% increase in relative eyelid aperture—a feature seen and encouraged by breed standards of brachycephalic breeds—were more than three times likely to suffer from corneal ulceration. Visible sclera was associated with almost three times the rate of corneal ulcers, while dogs with nasal folds were five times more likely to suffer from corneal ulcers. Affected dogs may require medical management, and/or surgery.

Brachycephalic breeds may experience a higher incidence of orthopaedic conditions. The French Bulldog is among the breeds with the highest incidences of patellar luxation, an orthopedic condition which can lead to lameness, osteoarthritis and pain [79]. The prevalence of patellar luxation was 4% in French Bulldogs, 3.8% in CKCS, 3.5% in Pugs and 2.9% in Bulldogs, compared with an overall prevalence of 1.3%. Affected dogs may require medical management, and/or surgery.

## 9. Behavioural Consequences of Canine Brachycephaly

Dogs vary in their morphology more than any other mammalian species. Because there is a wide spectrum of head shapes, with so many permutations of skull width and length [80], it may be simplistic to categorise animals as either brachycephalic or non-brachycephalic. Other terms have been used to identify non-brachycephalic breeds include mesocephalic (effectively mid-ranged skull) and dolichocephalic (long-skulled). However, it has been argued that even these three categories of skull shape are simply arbitrary, and that a sliding scale of cephalic index (CI, a measure of the ratio between skull width and length) is a more elegant means of studying the skull differences among dogs and their consequences. Breed groupings and genetic clusters only partially explain CI differences. This suggests that the three categories of canine skull (dolichocephalic, mesocephalic and brachycephalic) are overly arbitrary, and that CI is a better metric of this dimension than three categories, the ranges for which many authors disagree [81]. There is evidence of sexual dimorphism for head length in five breeds, head width in 10 breeds and CI in two breeds [82].

While fashion influences the popularity of certain breeds [83], the rise of brachycephalic dogs (and the concomitant increase in CI) may also reflect the behavioural appeal of some breeds [3]. That said, it is unclear how much behaviour has influenced breed demand. Of course, much of what veterinarians know about dog-keeping and dog behaviour is based on the assumption that dogs do not have extreme head-shapes. Unless one has lived with a brachycephalic dog, one may underestimate the effects a compromised airway can have on canine behaviour, welfare and quality of life.

Dogs with short skulls are prone to prominent facial skin folds. In addition to predisposing dogs to dermatitis and exposure keratitis [84], prominent facial skin folds may also affect the dog’s ability to lift its lips to signal fear and aggression to conspecifics, as has been described in neotenised (or paedomorphic) dogs [85]. That said, the extent of any social deficit associated with brachycephalism has yet to be defined. Other consequences of brachycephalism include crowded dentition, which may compromise the dog’s ability to chew; this may have consequences for stress reduction, since it has been suggested that chewing is a fundamental canine stress-coping mechanism [86].

If dogs’ physical ability to see, chew and signal effectively is affected by skull shape, so too is brain morphology. Regodon et al., [87] showed that, with reduced skull length in dogs, the cranium becomes more perpendicular relative to the facial axis and leads to an increase of structural disorders such as hydrocephaly [88] (see Section 3 above) and occipital dysplasia [89]. Midline sagittal MRI slices of canine brains across a broad range of CI (cephalic index) have revealed that cranial diversity is associated with systematic differences in brain morphology [90].

Increasing CI is associated with a progressive ventral pitching of the primary brain axis, as well as with a ventral shift in the olfactory lobe position [90]. This reflects an extraordinary degree of intra-species variability in brain anatomy that appears to be independent of brain size or bodyweight differences, and is primarily driven by human-driven selection pressure. The olfactory bulb of dogs with high CI seems to have migrated to a potential space ventral to the orbital frontal cortex, thus liberating the anterior pole for the normal development of the frontal cortex. In contrast, dogs with low CI may have capacity in the cranial vault to allow the olfactory bulb to develop almost directly anterior to the frontal lobe [90].

There is a strong correlation between the distribution of retinal ganglion cells and CI in dogs, which is likely to affect vision. This, in turn, may contribute to behavioural differences. The ganglion cells of dogs with a high CI are concentrated in the central circular area of the retina, which provides good visual acuity in the centre of the visual field [91]. In contrast, those of the dogs with a low CI are arranged in a linear fashion across the retina, to create a so-called visual streak that is thought to provide excellent peripheral vision. 

To explore the relationships between morphology and behaviour, data on breed-specific CI, height and bodyweight (based on a survey of >1000 dogs) have been integrated with owners’ reports of canine behaviour (*n* = 8301) across 49 different breeds [92]. Stepwise backward regression has revealed that across breeds, eight behavioural traits correlate with either CI alone (*n* = 3), bodyweight-and-skull shape combined (*n* = 2) or height-and-skull shape combined (*n* = 3). These results demonstrate that the canine head shape, and therefore brain morphology, co-varies with behaviour. Specifically, these results have shown that, compared with long-skulled dogs, short-skulled dogs are significantly more likely to show compulsive staring, self-grooming, allo-grooming and dog-directed aggression. In contrast, they are significantly less likely to display behaviours such as stealing food, stranger-directed fear, persistent barking and chasing [92]. Some of these traits may reflect the retinal and brain effects of a high CI, and may even begin to explain some of the charm and appeal of the currently popular short-skulled breeds. 

Brachycephaly may restrict the ability of dogs to exercise. Exercise intolerance reduces the dog’s ability to shed excessive weight but also to express its *telos* (or dogginess) (for a discussion of *telos*, see Rollin [93]. Brachycephalic dogs experience increasing respiratory effort at rest and likely air hunger, an unpleasant negative welfare state, during exertion [94]. Brachycephalic dogs are less capable than non-brachycephalic dogs of thermoregulating during periods of heat stress, with a significantly greater increase in respiratory rate as ambient temperatures rise, compared with healthy non-brachycephalic dogs [5]. In a study using whole-body plethysmography to continuously measure respiration as ambient temperature was increased, five of 52 brachycephalic dogs and none of 53 non-brachycephalic dogs, were withdrawn from heat treatment because they showed signs of respiratory distress [5]. The authors found that body condition score was positively associated with body temperature (independent of environmental conditions) and negatively associated with tidal volume, suggesting that overweight brachycephalic dogs are less tolerant of heat stress.

When assessed using airway auscultation pre- and post-exercise (a gentle three-minute trot), and whole-body barometric plethysmography, approximately 50% of Pugs and French Bulldogs, and 45% of Bulldogs had clinically significant airway disease [8]. A survey of owners of 61 Pugs and 39 French Bulldogs referred for surgical treatment of BOAS revealed that 88% of brachycephalic dogs had severe exercise intolerance and required prolonged recovery time following exercise, 56% experienced sleep problems, and 50% had significant sensitivity to heat [95]. The mean age of dogs at the time the survey was completed was 3.33 years (a range of eight months to 11 years), with the mean age of onset of signs being 1.12 years (a range of three months to four years). Many affected dogs had developed strategies to avoid airway obstruction during sleep, including adopting a sitting position while sleeping, elevating their chin, or sleeping with a toy between their teeth to keep the mouth open to compensate for nasal obstruction. According to the survey, 10% of dogs could sleep only with an open mouth.

Airway abnormalities, coupled with a propensity for heat stress, may prevent brachycephalic dogs from safely participating in household activities, including vacations. According to data collected by the US Department of Transport, of the 122 dogs that died while being transported as cargo in commercial aircraft between 2005 and 2010, approximately half were brachycephalic breeds, the English Bulldogs (25 deaths), Pugs (11 deaths) and French Bulldogs (six deaths) accounting for the highest numbers [96]. For this reason, many airlines have placed a ban or heavy restrictions on the transport of brachycephalic breeds. For example, Qantas does not permit the transport of pure-bred brachycephalic breeds, including Bulldogs (French or English), Pugs, Pekingese, Boston terriers and Japanese chin, on flights longer than five sectors, or more than two sectors per journey. If transported, these dogs must be contained in an approved crate that is twice the size of the minimum requirement for normal dogs. For international flights of over five hours, owners must submit a form of acknowledgement and indemnity [97].

In summary, there are many ways in which increasing CI may compromise the dog’s ability to behave as a normal dog would, and as dogs have evolved and been domesticated to. 

Canine and human co-evolution have disclosed remarkable morphological plasticity in dogs [98]. For the owners of French Bulldogs, high levels of problems are negatively associated with plans to acquire the same breed again [99]. This may have implications for the veterinary caseloads of the future. Whether the recent selection of dogs with progressively diminished CI has reached physiological limits is controversial [84]. The short life expectancy of dogs with a very high CI [7] suggests that we may have even exceeded these limits.

## 10. Ethical Challenges Associated with Brachycephalic Breeds

The brachycephalic patient, and indeed other patients with extreme phenotypes due to selective breeding, may place veterinarians in ethically challenging situations when approached to assist in the treatment and breeding of affected animals [100]. In drawing attention to BOAS, veterinarians may appear to be critical of the very features that clients find most endearing about their companion animals. 

In addition, veterinarians may fear that, in drawing an owner’s attention to the severity and chronicity of clinical signs that the dog has been and is experiencing, they may be perceived to be judging and therefore alienating clients, as well as the breeders of these animals. In addition, veterinarians may have a conflict of interest if they draw an income from treating BOAS. For example, writing anonymously in *The Guardian*, one veterinarian suggests that it is this conflict of interest which prevents veterinarians from speaking publicly about the negative health and welfare impacts of brachycephaly:
“…the vast majority of us work in general practice and our income is based on mending people’s animals and getting paid for it, and, like it or not, a large number of those clients have brachycephalic dogs. In my practice alone we have a number of pug, shih-tzu and bulldog breeders and dozens of owners with squashed-nosed pets…If I stood up and told the truth about these breeds, I would immediately alienate them and they would up sticks and move to the neighbouring practice where the vet was not as outspoken. Vets in general practice simply cannot afford to be honest and to speak out. You would be hard-pushed to find a general practitioner who likes the concept of a brachycephalic dog but you would be equally hard-pushed to find one being openly critical of them because this would put their livelihood on the line.”[101]


That author’s untested hypothesis is that being critical of brachycephalic breeds will alienate clients who are owners of brachycephalic dogs, potentially eliminating the opportunity for the veterinarian to have any influence on the dog’s health or welfare. A further implication is that in such cases, a veterinarian’s secondary interest, that is, drawing an income, takes primacy over their primary professional considerations—the interests of the animal and the client.

The response of veterinarians to brachycephalic patients depends on how they perceive their role. If they believe that their role begins and ends with treating the individual patient to the best of their ability, they can treat BOAS as a congenital disease, developing and refining their skills to provide the best possible outcome for each patient. In this narrow conception of the professional role, the stakeholders include the dogs (whose clinical signs are managed), the clients (whose feelings are dealt with sensitively) and the veterinarians (who may alleviate suffering and are paid for their efforts). Even within this conception, Hernandez and colleagues [102] argue that veterinarians have a professional duty to “speak up” about animal welfare issues, despite the risk of having uncomfortable conversations. Similarly, Coghlan [103] argues that companion animal veterinarians are ethically obliged to act as strong patient advocates, in that they must first consider the interests of the patient they are treating. To ensure appropriate and timely treatment, and to optimize welfare, it is in the interests of the patient for the client to appreciate the degree of suffering that is associated with BOAS, the pathophysiology of the condition, and steps to take to minimise suffering and to improve health and welfare. 

However, it can be argued that veterinarians have obligations extending beyond the confines of the consultation room. For example, in New South Wales, veterinarians swear an oath to “practice veterinary science ethically and conscientiously for the benefit of animal welfare, animal and human health, users of veterinary services and the community” [104]. Similarly, veterinarians in the UK swear to uphold their responsibilities to “the public, my clients, the profession, and the Royal College of Veterinary Surgeons” [104]. Such oaths suggest that simply serving the interests of the client, the individual patient and themselves and/or employer abrogates veterinarians’ professional responsibilities to the wider animal population and community. 

As stated in the British Veterinary Association’s Animal Welfare Strategy, pointedly titled “Vets Speaking Up for Animal Welfare”:

“Reputationally, if [vets] don’t speak out about systemic animal welfare problems or if we only do so reactively once a critical mass of favourable public opinion has been achieved, then this can lead to accusations of weak morality and, worse, complicity in animal welfare problems” [105].

The failure of veterinarians to speak up about welfare problems that are associated with selective breeding is cited as an example. If, as the Strategy suggests, our duty as professionals extends beyond the individual vet–client–patient relationship, that is, promoting animal welfare on a broader scale, we are obliged to do what we can to optimise the welfare of the individual animal presented to us, as well as evaluate and act on systemic animal welfare issues at the level of the surrounding community, and through professional associations “to achieve political impact and challenge societal norms” [105].

Veterinary oaths and codes of conduct hold that veterinarians should be of good character. In terms of ethical theories, this is aligned with virtue ethics. Virtues are morally relevant, reliable character traits that we expect in persons of good character.

Beauchamp and Childress [106] identify care, as well as five focal virtues, as being important for medical professionals, which may be applied in relation to veterinarian’s obligations towards brachycephalic dogs, as outlined in Table 2.

Consequentialism is a category of ethical theories that judge the rightness or wrongness of an action by its consequences. Utilitarianism is a branch of consequentialism that determines the rightness or a wrongness of an action by its ability to maximise well-being, or to minimize suffering.

There is a real concern that, in failing to speak publicly about the health and welfare impacts of brachycephaly and other breed-associated conditions, veterinarians perpetuate the problem. In a UK study, 58% of owners of dogs with BOAS reported that their dog did not have a breathing problem, despite reporting a high frequency and severity of clinical signs [107]. This ignorance is perpetuated if veterinarians accept signs of BOAS as being normal for the breed [8]. Without an appreciation of the health and welfare implications of BOAS, owners may not seek treatment for affected dogs. Furthermore, potential owners may select affected dogs as companion animals, and affected dogs may continue to be used for breeding. Such consequences would increase overall suffering. One study found that the experience of health and behaviour problems, including breed-related problems, while their dogs did not impact the owner’s likelihood of adopting a dog of the same breed again [99]. The one exception was French Bulldogs, where the extent of health problems did decrease the owner’s intention to acquire the same breed again. This may be because appearance is a stronger influential factor in choosing a dog than health and longevity. For example, despite an increase in media coverage of the link between conformation and health problems, and the increased availability of educational resources for potential puppy buyers, a UK study found that persons who purchased a brachycephalic dog prioritized appearance over health and longevity [108].

Because of the increasing popularity of brachycephalic breeds, BOAS affects larger numbers of animals than ever before, and it has the potential to continue to do so, due to its association with brachycephalic conformation. Furthermore, it may have severe adverse effects on affected animals for extended periods [107]. Again, this increases overall suffering.

Utilitarian frameworks hold that we should aim to minimise costs and maximise benefits. One of the best-known utilitarian frameworks is the 3Rs of replacement, reduction and refinement, used to assess proposed animal experiments [109,110]. According to this framework, animals may be used in experiments only if welfare costs are minimised whilst scientific benefits are maximised. The benefits must not be trivial [110]. When it comes to breeding brachycephalic dogs, it could be argued that the animals bear most of the welfare costs, while the benefits to humans of a fashionable animal are trivial. According to this framework, if animals are to be “used” as companions, welfare costs or harms should be minimised to the greatest degree possible. 

As has been argued by Rollin, our predominant social ethic combines utilitarianism with deontology, a category of ethical frameworks that judge rightness and wrongness against moral rules. It has been argued that the “five freedoms of animal welfare” (freedom from hunger and thirst; discomfort; pain, injury and disease; fear and distress; and freedom to express normal behaviour) function as rights [111]. In other words, it is generally held that, at a minimum, animals that are kept and used by humans are afforded such freedoms.

The previous discussion demonstrates that dogs with BOAS do not enjoy freedom from discomfort, nor freedom from pain, injury and disease, and they do not enjoy the freedom to express normal behaviour. According to both deontological and utilitarian ethical frameworks, therefore, the breeding of dogs with extreme phenotypes, including BOAS, cannot be justified, and further, it must be discouraged because of our duties to companion animals. 

## 11. The Veterinarian’s Role

Veterinarians are responsible for the health and welfare of individual animals, but are also expected to be involved in policy discussions regarding health and welfare, including breeding practices [112]. The veterinarian can address the issue of extreme brachycephaly at the level of the patient, client, breeder, community and politically.

As discussed, at the level of the patient, the veterinarian must be able to identify and treat suffering. This requires an understanding of the pathophysiology of BOAS, as well as the welfare implications. When advising owners about selecting a companion animal, veterinarians can educate them about the health and welfare risks of extreme morphology and inherited disorders, as well as potential financial implications, and recommend breeds that are fit for function. When presented with a brachycephalic patient, the veterinarian should identify and explain all brachycephaly-related clinical signs and recommend early intervention for animals at risk of BOAS (or refer appropriately). The veterinarian can recommend lifestyle changes for animals at risk of BOAS, including managing food intake, maintaining an appropriate body condition score, avoiding over-exertion and minimising exposure to high temperatures and high humidity [8]. Veterinarians can perform surgery and treatment, as described in Section 5, to improve patient breathing or to address specific problems such as lagopthalmos (inability to blink completely) [78]. However, Packer argues that in the longer term, veterinarians can do more than simply surgically intervene to return individual dogs to a lower risk morphology [78].

At the level of the client, the veterinarian should be frank and honest, providing an evidence-based explanation of concerns. Materials such as those produced for the joint AVA (Australian Veterinary Association) and RSPCA (Royal Society for the Prevention of Cruelty to Animals) Australia Love is Blind campaign may be shared with clients http://www.ava.com.au/loveisblind/resources-vets. Practicing sensitive, open communication around a range of issues (from breaking bad news, discussing companion animal obesity and communicating frankly about terminal or chronic conditions) is important. It is acceptable for a professional to make strong statements about animal welfare. Veterinarians are well placed to advise pet owners on breed-related health and welfare issues. However, many pet owners acquire their pets prior to visiting a veterinarian, so that many relevant conversations occur post-purchase. Breed-related conditions, the likely associated costs and the welfare impact on the pet in the short- and long-term should be discussed as early as possible with clients. 

Where breeders of extreme brachycephalic animals are clients, it is important to help them to improve the health of their animals. This can be achieved by strongly recommending the neutering of animals with BOAS, or those requiring caesarean sections, and stating that this recommendation is based on health and welfare grounds. Veterinarians may refuse to participate in artificial insemination and other breeding services, if breeders are unwilling to address the health issues of their breeding stock. Veterinarians may also carry out exercise tolerance tests and grading of brachycephaly and participate in health schemes in conjunction with kennel clubs [2]. Practices can enrol in clinical surveillance programs such as VetCompass to contribute to the clinical evidence base, to facilitate monitoring and management of inherited conditions. 

At the level of the community, veterinarians can promote pre-purchase consultations to prospective owners, and outline health problems associated with brachycephalic conformation. In addition, they can avoid posting brachycephalic breeds on social media and marketing material. Veterinarians can support breeders and breed associations that are determined to genuinely improve the health of welfare of these dogs.

At the level of the profession, veterinarians can take an active role in their professional association by working with kennel clubs to promote the need for change in breed standards, and to be involved in initiatives to address the health and welfare of breeds [105], including their involvement in organisations such as the UK’s Advisory Council on the Welfare Issues of Dog Breeding and the International Partnership for Dogs. Some breeds may not retain enough genetic diversity to permit the correction of the extreme brachycephalic phenotype and its associated health problems. A genetic assessment of the English Bulldog identified low genetic diversity, concluding that the possibility of using reverse selection to improve the health of purebred English Bulldogs, select against deleterious simple recessive traits, and/or accommodating further genetic manipulations without further decreasing genetic diversity was questionable [74]. The authors concluded that judicious outcrossing may be required to address these concerns. Such measures should be investigated and discussed with kennel clubs. 

Professional associations can educate the wider public by providing education programs and indirectly, by not promoting extremely brachycephalic breeds. In 2017, the British Veterinary Association announced that it would no longer use images of brachycephalic breeds in advertising to prevent normalisation of their associated health issues and to reduce demands for these breeds [2,113]. The AVA took a similar position the same year, also avoiding the use of images of other breeds with exaggerated features, such as Shar-peis and Dachshunds [114]. In addition, professional associations can lobby for strengthened animal welfare legislation. A number of countries, including Austria, Germany and Switzerland [115,116,117], have introduced “anti-qualzucht” (“torture breeding”) clauses in animal welfare legislation, which on some interpretations, makes it illegal to knowingly breed animals that are likely to experience pain, distress or harm (for example, shortness of breath). Because only a small percentage of the canine brachycephalic population is registered, efforts must be made to crack-down on irresponsible backyard breeders and puppy farms that are focused on profits over welfare [8]. This may require changes in legislation around the breeding of dogs.

In January 2018, New Zealand auction and classified advertising website Trade Me (www.trademe.co.nz) announced a ban on the sale of Pugs, British Bulldogs and French Bulldogs, and their crossbreeds, on the grounds that breeders cannot claim puppies are 100 per cent healthy due to the high risk of BOAS in these breeds [118]. In an explanation posted on its website, the company stated that while it is likely that the animals will be sold elsewhere online, “Trade Me does not want to contribute to this problem” and sees its stance as “an opportunity to educate potential buyers”, and called for the Government to consider regulatory intervention to address BOAS in these breeds [118]. Veterinarians and associations were consulted prior to the ban.

In summary, veterinarians have a professional and moral obligation to work to prevent and minimise the negative health and welfare impacts of extreme morphology and inherited disorders by working at the level of the patient, their communities and with breed and professional associations. Because of the health and welfare costs of BOAS and other conditions associated with the extreme brachycephalic phenotype, coupled with the growing popularity of these breeds, veterinarians must take a strong position against perpetuating these phenotypes.

## 12. Conclusions

Breeding of dogs with extreme brachycephaly is associated with health and welfare problems, including acute and chronic respiratory distress, with the potential need for airway stabilization and surgical intervention to manage BOAS. In addition, brachycephalic breeds present an anaesthetic challenge. Extreme brachycephalic confirmation is associated with non-airway related conditions, including neurological, dermatological, ophthalmic and orthopedic conditions. In addition to impacting the health and welfare of these dogs, owners may incur significant costs in treating them. The growing popularity of these breeds suggests that companion animal owners do not appreciate the enormity of these health and welfare costs. Veterinarians have a professional and moral obligation to prevent and minimise the negative health and welfare impacts of extreme morphology and inherited disorders and must address extreme brachycephalic conformation, not only at the level of the patient, but also as a systemic welfare problem. 

## Figures and Tables

**Figure 1 animals-09-00003-f001:**
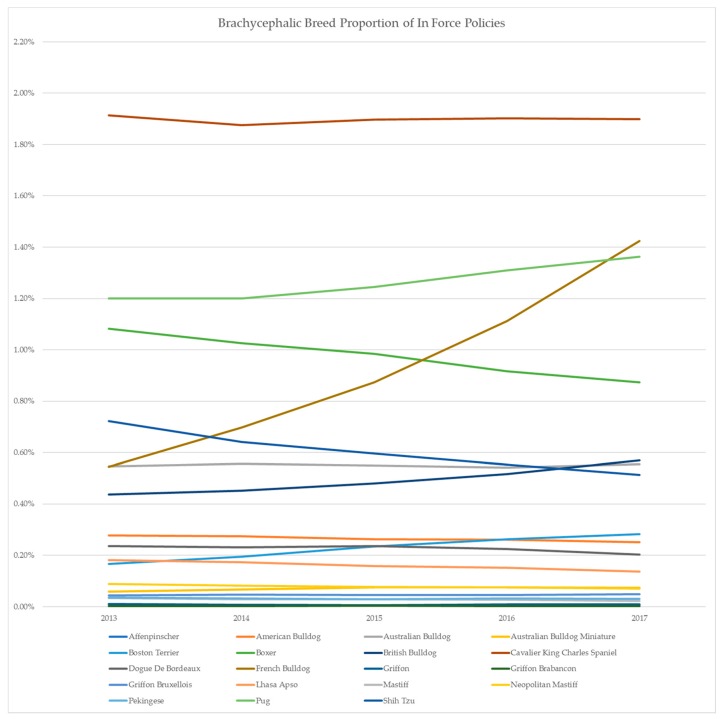
Distribution of PetSure insurance policies in force (2013–2017) for Boxers, British Bulldogs, French Bulldogs and Pugs.

**Figure 2 animals-09-00003-f002:**
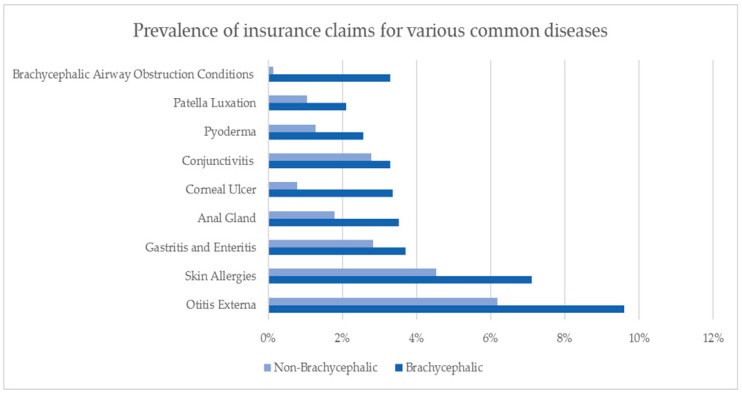
The prevalence of PetSure insurance claims relating to common diseases in brachycephalic and non-brachycephalic breeds from 2013–2017.

**Table 1 animals-09-00003-t001:** Annual percentage changes in popular breeds insured by PetSure Australia. These numbers represent the growth for each breed as a proportion of all policies in force for each calendar year.

Breed	Year on Year Growth			
	2014	2015	2016	2017
Affenpinscher	−19.29%	−17.65%	−3.60%	35.59%
American Bulldog	−0.91%	−4.33%	−0.95%	−3.97%
Australian Bulldog	1.70%	−1.23%	−1.51%	2.42%
Australian Bulldog Miniature	14.45%	13.29%	0.98%	−3.05%
Boston Terrier	17.04%	20.04%	12.02%	7.38%
Boxer	−5.24%	−3.91%	−6.89%	−4.73%
British Bulldog	3.35%	6.05%	7.81%	10.21%
Cavalier King Charles Spaniel	−2.05%	1.20%	0.27%	−0.20%
Dogue De Bordeaux	−2.43%	2.46%	−4.78%	−10.08%
French Bulldog	28.00%	25.04%	27.44%	28.08%
Griffon	−30.29%	−17.65%	41.56%	7.23%
Griffon Brabancon	13.00%	29.40%	−13.24%	−4.57%
Griffon Bruxellois	6.75%	−2.92%	0.52%	4.66%
Lhasa Apso	−5.38%	−8.67%	−3.31%	−9.84%
Mastiff	−15.92%	−2.28%	−5.44%	−15.23%
Neopolitan Mastiff	−7.57%	−6.83%	−1.36%	−5.82%
Pekingese	−10.56%	−7.61%	7.51%	−1.78%
Pug	−0.01%	3.70%	5.25%	4.07%
Shih Tzu	−11.28%	−7.01%	−7.40%	−7.18%
Total	0.00%	2.53%	3.06%	3.85%

**Table 2 animals-09-00003-t002:** Beauchamp and Childress’ focal virtues for medical professionals, and how these may manifest in the veterinarian presented with the brachycephalic patient.

Virtue	Manifestation
Care	The veterinarian has an emotional commitment to, and the willingness to act on behalf of persons and patients.
Compassion	The veterinarian has an active regard for both the animal and owner’s welfare, with imaginative awareness and sympathy, tenderness and discomfort at another’s suffering. The ability to identify and motivation to address suffering.
Discernment	The veterinarian is able to make appropriate judgements and decisions without undue influence of fears, personal attachments or inducements.
Trustworthiness	The veterinarian can be trusted to give an honest, informed opinion about the patient’s condition, potential causes and contributing factors, and prognosis, and to declare any conflicts of interest.
Integrity	The veterinarian is faithful to his or her moral values, and will defend these when necessary.
Conscientiousness	The veterinarian works conscientiously to do what is right: to provide the best possible care to the individual patient, and to future patients by remaining up-to-date with scientific evidence. The conscientious veterinarian strives to prevent disease at the level of the individual, as well as that of the population.

Adapted from [106] (pp. 37–44).

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
