# Peer review of "Consequences and Management of Canine Brachycephaly in Veterinary Practice: Perspectives from Australian Veterinarians and Veterinary Specialists"

_animals, 2018, doi:10.3390/ani9010003_

Round 1
Reviewer 1 Report
Overall a nice review of the current status quo surrounding brachycephalic breeds with some useful international elements. The structure and choice of issues to discuss is a little hard to follow at present. My main suggestion would be to either more thoroughly justify the disorders you focus on or omit that are common in brachycephalic dogs, or add additional sections on the most common/severe disorders that are currently not included but may be of high relevance to welfare. The inclusion of original data (if the PetSure data are?) should be described more thoroughly to ensure their integrity and understand their context. With some edits and inclusions this will be a useful snap-shot of the state of knowledge on this topic and the vets role within it in 2018.
Title: As this review a veterinarian’s perspective, I think having reference to this in the title would be useful (also to make it clear it is a review rather than original research paper to a reader skimming the title)
Introduction: Useful to open with a definition of what you mean by a brachycephalic dog and include some lay language for a non-technical reader
Does the Australian KC publish registration stats that you could calculate % increase in popularity from, to allow direct comparison with the UK data you present?
Small point but keep consistent in the capitalisation of breed names
If this is the first time these PetSure data have been presented then it would be worth highlighting these findings in your abstract to signpost potential readers to them. Some additional data to describe the insurance database population they are derived from would also be valuable for epidemiological context
In terms of presentation of health issues and their treatment in clinical practice, as much of your intro focuses on airways I would have expected section 4 to come before your section on brain disorders in these breeds
Why have you focused on some conformation related disorders in brachys but neglected others? There is a growing literature base on spinal malformations in brachy breeds that are common and have the potential to severely impinge welfare; for completeness I would suggest their inclusion here to provide a more comprehensive overview of the brachy CNS.
On that line – ophthalmological and dermatological issues are prevalent in many of the breeds you mention, is there a reason for not including them to some extent here?
Some of section 7 is not relevant to behaviour (e.g. enzyme function) so it may be worth having a separate section on other physiological differences of unknown consequence
Ln538 this study only investigated French Bulldogs as an exemplar brachycephalic breed, so re-wording to reflect this more accurately would be useful – can only speculate on other brachy breeds at present
Ln571-585 I think this section on farm animals detracts from the well-made point about conflicts in small animal practice regarding brachy breeds and weakens this section, so I would suggest you remove these paragraphs/quote
Ln642 also data showing that brachy owners do not prioritise breed health or longevity in breed selection compared to non-brachy owners that may be of relevance here:
Packer RMA; Murphy D; Farnworth MJ (2017) Purchasing popular purebreds: Investigating the influence of breed-type on the pre-purchase attitudes and behaviour of dog owners. Animal Welfare 26: 191-201
There are now several brachycephalic working groups across Europe that are bringing together vets and other key stakeholders to tackle brachycephalic health that would be timely to mention here, along with international initiatives such as the International Partnership for Dogs bringing together kennel clubs on this issue
If you are going to mention legislation then a more thorough international overview would be valuable, for example what about 'Qualzucht' legislation in continental Europe e.g. Germany?
Reviewer 2 Report
Other than the length of this article which makes it hard to 'wade through', this is yet another comprehensive study of the disorders that face pet guardians, breeders and the veterinary community when caring for the ever-popular brachycephalic dog breeds. It would be helpful to include a list of all abbreviations used (other than obvious ones) up front.
The section on the role of veterinarians in speaking out about the ethical concerns of breeding these structurally abnormal dogs is compelling -- we must do so!
Three other comments: 1) The authors should include more reference to the parallel very large study conducted in the UK and published in 2015 (reference 79) and need to introduce it in the opening Introduction. 2) Figure 2 is very helpful but does not include respiratory conditions. Surely, these are important and should be added, or at least described from that data base. 3) Confused about why some references are cited in the text by author(s) and year whereas the majority are given numbers in the text, as usual. Was this an oversight?
Author Response
Please see attached word document

Reviewer 3 Report
The document would benefit greatly from being reorganized and rewritten based on one or a few specific objectives so that its content flows smoothly.
In a number of sections, the text lists a series of articles without a clear argument being developed. The introduction, for example, list in detail a number of studies without providing clear objectives. The reader is wondering whether the goal of the study is solely to increase awareness (of what: how debilitating is brachycephaly? How important it is to advise owner against owning dogs with chronic health problems?) among care providers.
Based on the text, this reviewer thinks (but could be mistaken) that the main goal of the authors appears to be to build an argument about the fact that veterinarian should become more engaged in providing advice regarding the ownership of a dog with a chronic health problem. Most of the sections in the article have little to do with the main objective of the authors. The fact that brachycephalic dogs are becoming more popular has little relevance, for example. The fact that they are difficult to anesthetize is barely relevant. Is the problem really growing or are we becoming more aware of it? Information about awareness among veterinarians regarding the health consequences of brachycephaly in dogs is lacking. Information about the decision making of owners when selecting a dog is lacking. Information about the current behaviour of veterinarian (with regard to providing advice regarding conditions leading to chronic pain or disability in dogs) or changes in that behaviour over time is also lacking. Are veterinarians in agreement with the authors but owners do not listen? Are veterinarians unaware? Unwilling to discuss? Willing to discuss but not able to discuss well? Are veterinarians able to discuss well but not sufficiently trusted or influential?
The section “Australian insurance data” should not be an independent section. It is only there to make the point that the number of brachycephalic dog (that are insured) is increasing. That point is barely relevant to the study. Also that point should be made more specifically. Total number of dogs, number in all brachycephalic breeds, etc. must be provided.
As another example, the first two paragraphs in the section “Australian insurance data” have nothing to do with Australian insurance data. The first paragraph discusses the fact that veterinarian offer advice to dog owners (without stating specifics about where prospective dog owners get advice or how prospective dog owners make decisions regarding getting a dog). The second paragraph discusses a British TV programme on dog health. These paragraphs do not belong in this section and the section does not really belong in the manuscript.
As another example (line 489), how is having wrinkled skin relevant to a section aimed at discussing the behaviour of brachycephalic dogs?
“Australian insurance data”. You focus on data from a single insurance company. Please bring perspective to this information: Are there no other insurance companies? How many dogs are insured? Is the number of dogs in Australia increasing? Is the number of insured dogs increasing?
The paragraph list does not appear to follow much logical progression:
Insurance data
Neurologic problems (why list before respiratory problems, since the respiratory problems are most common and most impactful?)
Respiratory problems
Sedation and anesthesia (this section belongs within respiratory problems)
Surgical management of respiratory problems (this section belongs within respiratory problems)
Behaviour
Ethics
Veterinarians’ roles
Sections should be organized based on the objectives of this article. If the goal is to review the medical issues faced by brachycephalic dogs, these should be reviewed in specific order.
If the purpose is to discuss the ethics of owning a dog with chronic conditions induced by breeding practices, the text should focus more on the ethics of owning dogs with these conditions
Why are orthopedic problems not discussed. Figure 2 suggests that their rate is much higher (at least for patellar luxation) than non brachycephalic breeds. How about hip dysplasia, the most common orthopedic disease?
Figure 1. Please show more than 4 breeds. You list 21 breeds in the text (lines 137 to 141). The data should be listed for all 21 breeds with percentage changes for all of them and as a total.
Figure 1. Please define what the ‘percentage’ in the y axis means, is it the percentage of dog insured or the percentage of dogs in the population?
Figure 1. How are the raw number of dogs changing, surely, the number of dog insured is increasing also? These data must be reported. I could envision some for of bias leading to an increase in the frequency of brachycephalic dogs being insured (relative to non-brachycephalic dogs) without an actual increase in the number of brachycephalic dogs being in existence (for example if owners of brachycephalic dogs were more sensitized to the health problems of their dogs than owners of non-brachycephalic dogs).
Figure 1. Where is the purported increase in frequency of brachycephalic dogs coming from / which breeds are getting less popular? How have numbers changed in non-chondrodystrophic breeds, please provide some information about non-brachycephalic breeds, maybe the 5 most popular about non-brachycephalic breeds?
Figure 1. The gray lines cannot be distinguished from each other. Please make lines easier to identify or (better) create a Table with data for all 21 breeds, etc.
Line 153. You speculate without much attention to the facts. Please provide the following information: number of dogs with mild/moderate brachycephaly in 2013 and in 2017, number of dogs with severe brachycephaly in 2013 and in 2017 (and proportion of overall brachycephalic dogs. The change in the number of surgical claims could also be due to a change in practice approaches. Veterinarians may have changed their threshold and tolerance for respiratory disorders in dogs. This could be investigated in a survey.
Line 193. The acronym CKCS should be introduced line 140 (first time the term is written in the text).
Author Response
Please see attached word document
